# Effective Meta-Regularization by Kernelized Proximal Regularization

**Weisen Jiang[1, 2], James T. Kwok[2], Yu Zhang[1, 3]** [*]
[1] Department of Computer Science and Engineering, Southern University of Science and Technology
[2] Department of Computer Science and Engineering, Hong Kong University of Science and Technology
[3] Peng Cheng Laboratory
{wjiangar, jamesk}@cse.ust.hk, yu.zhang.ust@gmail.com

## Abstract

We study the problem of meta-learning, which has proved to be advantageous to accelerate learning new tasks with a few samples. The recent approaches based on deep kernels achieve the state-of-the-art performance. However, the regularizers in their base learners are not learnable. In this paper, we propose an algorithm called MetaProx to learn a proximal regularizer for the base learner. We theoretically establish the convergence of MetaProx. Experimental results confirm the advantage of the proposed algorithm.

## 1 Introduction

Humans, by leveraging prior knowledge and experience, can easily learn new tasks from a handful of examples. In contrast, deep networks are data-hungry, and a large number of training samples are required to avoid overfitting. To reduce the labor-intensive and time-consuming process of data labeling, *meta-learning* (or *learning to learn*) [3, 37] aims to exact meta-knowledge from seen tasks to accelerate learning on unseen tasks. Recently, meta-learning has been receiving increasing attention due to its diverse successful applications in few-shot learning [41, 12, 39, 35], hyperparameter optimization [14], neural architecture search [24, 44], and reinforcement learning [29].

Many meta-learning algorithms operate on two levels. A base learner learns task-specific models in the inner loop, and a meta-learner learns the meta-parameter in the outer loop. A popular class of algorithms is based on meta-initialization [12, 25, 11, 38], such as the well-known MAML [12]. It learns a model initialization such that a good model for an unseen task can be learned from limited samples by a few gradient updates. However, computing the meta-gradient requires back-propagating through the entire inner optimization path, which is infeasible for large models and/or there are many gradient steps. During testing, it is common for MAML's base learner to perform many gradient steps to seek a more accurate solution [12]. However, for regression using a linear base learner and square loss, we will show that though the meta-learner can converge to the optimal meta-initialization, the base learner may overfit the training data at meta-testing.

Another class of meta-learning algorithms is based on meta-regularization [28, 43, 7, 8, 9], in which the base learner learns the task-specific model by minimizing the loss with a proximal regularizer (a biased regularizer from the meta-parameter). Denevi et al. [7] uses a linear model with efficient closed-form solution for the base learner. However, extending to nonlinear base learners requires computing the meta-gradient using matrix inversion, which can be infeasible for deep networks [28].

To introduce nonlinearity to the base learner, a recent approach is to make use of the kernel trick. For example, R2D2 [4] and MetaOptNet [22] use deep kernels [42] in meta-learning for few-shot

---

[*]Corresponding author.

classification. Specifically, the deep network is learned in the meta-learner, while a base kernel is used in the base learner. Though they achieve state-of-the-art performance, their base learners use a Tikhonov regularizer rather than a learnable proximal regularizer as in meta-regularization methods.

As learning a meta-regularization has been shown to be effective in linear models for regression [7] and classification [8], in this paper we propose a kernel-based algorithm to meta-learn a proximal regularizer for a nonlinear base learner. After kernel extension, the learnable function in the proximal regularizer is a function in the reproducing kernel Hilbert space (RKHS) induced by the base kernel. The proposed algorithm is guaranteed to converge to a critical point of the meta-loss and its global convergence is also established. Experiments on various benchmark regression and classification datasets demonstrate the superiority of the proposed algorithm over the state-of-the-arts.

**Notations.** Vectors are denoted by lowercase boldface, and matrices by uppercase boldface. For a vector $\mathbf{x}$, $\|\mathbf{x}\| = \sqrt{\sum_i x_i^2}$ and $\mathrm{diag}(\mathbf{x})$ constructs a diagonal matrix with $\mathbf{x}$ on the diagonal. $\|\cdot\|_{\mathcal{H}}$ is the norm on the Hilbert space $\mathcal{H}$. $\mathcal{N}(0, \sigma^2)$ is the univariate normal distribution with mean zero and variance $\sigma^2$. $\mathcal{N}(\mathbf{m}, \boldsymbol{\Sigma})$ is the multivariate normal distribution with mean $\mathbf{m}$ and covariance matrix $\boldsymbol{\Sigma}$.

## 2 Related Work

In meta-learning, a collection $\mathcal{T}$ of tasks are used to learn a meta-parameter $\boldsymbol{\theta}$ and base learner's parameters $\{\mathbf{w}_1, \ldots, \mathbf{w}_{|\mathcal{T}|}\}$. Each task $\tau$ is sampled from a given distribution $p(\tau)$, and has a support set $S_\tau = \{(\mathbf{x}_i, y_i) : i = 1, \ldots, n_s\}$ and a query set $Q_\tau = \{(\mathbf{x}_i, y_i) : i = 1, \ldots, n_q\}$, where $\mathbf{x} \in \mathbb{R}^d$ are the features and $y$ the labels. Let $f(\cdot; \mathbf{w})$ be a model parameterized by $\mathbf{w}$ and $\mathcal{L}(D; \mathbf{w}) \equiv \sum_{(\mathbf{x}, y) \in D} \ell(f(\mathbf{x}; \mathbf{w}), y)$ be the supervised loss on data set $D$, where $\ell(\cdot, \cdot)$ is a loss function that is assumed to be convex w.r.t. the first argument. In each meta-training iteration, a batch $\mathcal{B}$ of tasks are randomly sampled from $\mathcal{T}$. The base learner takes a task $\tau$ from $\mathcal{B}$ and the meta-parameter $\boldsymbol{\theta}$ to build the model $f(\cdot; \mathbf{w}_\tau)$. After all tasks in the batch are processed by the base learner, the meta-learner minimizes the loss $\sum_{\tau \in \mathcal{B}} \mathcal{L}(Q_\tau; \mathbf{w}_\tau)$ w.r.t. $\boldsymbol{\theta}$, and the iteration repeats. During meta-testing, given an unseen task $\tau' \sim p(\tau)$, a model $f(\cdot; \mathbf{w}_{\tau'})$ is similarly learned from $S_{\tau'}$ and $\boldsymbol{\theta}$. Finally, its performance is evaluated on $Q_{\tau'}$.

Popular meta-learning algorithms usually construct the task-specific model by: (i) meta-initialization [12, 25, 11, 38], (ii) meta-regularization [7, 8, 9, 28, 43], or (iii) metric learning [35, 4, 22, 27]. A representative meta-learning algorithm based on meta-initialization is MAML [12]. Its base learner learns $\mathbf{w}_\tau$ by gradient descent from a learnable initialization. Computing the meta-gradient $\nabla_{\boldsymbol{\theta}} \mathcal{L}(Q_\tau; \mathbf{w}_\tau)$ needs to back-propagate through the inner optimization path and involves second-order derivatives. This can be expensive for large models and/or when there are many gradient steps.

For meta-learning algorithms based on meta-regularization, the base learner learns $\mathbf{w}_\tau$ by minimizing a proximal regularized loss

$$\mathcal{L}(S_\tau; \mathbf{w}) + \frac{\lambda}{2}\|\mathbf{w} - \boldsymbol{\theta}\|^2, \tag{1}$$

where $\lambda > 0$ is the regularization parameter. The meta-gradient can be computed directly from the learned $\mathbf{w}_\tau$, without back-propagating through the inner optimization trajectory [28]. Though more efficient, this still takes $\mathcal{O}(n_{\boldsymbol{\theta}}^3)$ time [28], where $n_{\boldsymbol{\theta}}$ is the number of parameters in $\boldsymbol{\theta}$.

Metric learning methods have been widely studied in few-shot learning [39, 35, 4, 22, 27]. The meta-learner maps raw samples to an embedding space via a a deep network, then feeds the embeddings to the base learner to train a simple task model. Typical models include non-parametric prototype classifier (ProtoNet [35]), linear models like ridge regression (R2D2 [4]), SVM classifier (MetaOptNet-SVM [22]), and softmax classifier (ANIL [27]). In particular, the base learners in R2D2 [4], MetaOptNet [22] and DKT [26] seek solutions in the dual space, which achieve state-of-the-art performance. However, their base learners use a Tikhonov regularizer, which is not learnable as in meta-regularization approaches.

## 3 Meta-Regularization by Kernelized Proximal Regularization

### 3.1 Meta-Initialization versus Meta-Regularization

In this section, we consider a simple regression setting with linear model and square loss. Each task $\tau$ is a linear regressor with parameter $\mathbf{w}_\tau^* \in \mathbb{R}^d$. We assume that each input $\mathbf{x}$ is sampled

| **Algorithm 1** MAML [12]. | **Algorithm 2** CommonMean [7]. |
|---|---|
| **Require:** step size $\gamma$ and $\eta_t$, batch size $b$; | **Require:** hyperparameter $\lambda$, step size $\eta_t$, batch size $b$; |
| 1: **for** $t = 1, 2, 3, \ldots$ **do** | 1: **for** $t = 1, 2, 3, \ldots$ **do** |
| 2:    sample a batch $\mathcal{B}_t$ of tasks from $p(\tau)$; | 2:    sample a batch $\mathcal{B}_t$ of tasks from $p(\tau)$; |
| 3:    base learner: | 3:    base learner: |
| 4:    **for** $\tau \in \mathcal{B}_t$ **do** | 4:    **for** $\tau \in \mathcal{B}_t$ **do** |
| 5:       $\mathbf{w}_\tau^{(\text{gd})} = \boldsymbol{\psi}_t - \gamma \mathbf{X}_\tau^\top (\mathbf{X}_\tau \boldsymbol{\psi}_t - \mathbf{y}_\tau)$; | 5:       $\mathbf{w}_\tau^{(\text{prox})} = \min_{\mathbf{w}} \frac{1}{2}\|\mathbf{X}_\tau \mathbf{w} - \mathbf{y}_\tau\|^2 + \frac{\lambda}{2}\|\mathbf{w} - \boldsymbol{\theta}_t\|^2$; |
| 6:       $\mathbf{g}_\tau = \frac{1}{2} \sum_{(\mathbf{x},y) \in Q_\tau} \nabla_{\boldsymbol{\psi}_t}(\mathbf{x}^\top \mathbf{w}_\tau^{(\text{gd})} - y)^2$; | 6:       $\mathbf{g}_\tau = \frac{1}{2} \sum_{(\mathbf{x},y) \in Q_\tau} \nabla_{\boldsymbol{\theta}_t}(\mathbf{x}^\top \mathbf{w}_\tau^{(\text{prox})} - y)^2$; |
| 7:    **end for** | 7:    **end for** |
| 8:    meta-learner: $\boldsymbol{\psi}_{t+1} = \boldsymbol{\psi}_t - \frac{\eta_t}{b}\sum_{\tau \in \mathcal{B}_t} \mathbf{g}_\tau$; | 8:    meta-learner: $\boldsymbol{\theta}_{t+1} = \boldsymbol{\theta}_t - \frac{\eta_t}{b}\sum_{\tau \in \mathcal{B}_t} \mathbf{g}_\tau$; |
| 9: **end for** | 9: **end for** |

from $\mathcal{N}(\mathbf{0}, \sigma_{\mathbf{x}}^2 \mathbf{I})$, and the output $y$ is obtained as $\mathbf{x}^\top \mathbf{w}_\tau^* + \xi$, where $\xi \sim \mathcal{N}(0, \sigma_\xi^2)$ is the random noise. We compare two representative meta-learning algorithms: (i) MAML [12], which is based on meta-initialization and performs one gradient descent step in the inner loop of the bilevel optimization problem; and (ii) learning around a common mean (denoted CommonMean) [7], which is based on meta-regularization. It learns the model parameters for task $\tau$ by minimizing the loss with a proximal regularizer around the meta-parameter $\boldsymbol{\theta}$:

$$\mathbf{w}_\tau^{(\text{prox})} = \underset{\mathbf{w}}{\text{argmin}} \sum_{(\mathbf{x}_i, y_i) \in S_\tau} \frac{1}{2}(\mathbf{w}^\top \mathbf{x}_i - y_i)^2 + \frac{\lambda}{2}\|\mathbf{w} - \boldsymbol{\theta}\|^2 = (\lambda \mathbf{I} + \mathbf{X}_\tau^\top \mathbf{X}_\tau)^{-1}(\lambda \boldsymbol{\theta} + \mathbf{X}_\tau^\top \mathbf{y}_\tau), \quad (2)$$

where $\mathbf{X}_\tau = [\mathbf{x}_1^\top; \ldots; \mathbf{x}_{n_s}^\top]$ is the sample matrix from $S_\tau$, and $\mathbf{y}_\tau = [y_1; \ldots; y_{n_s}]$ is the corresponding label vector. Algorithms 1 and 2 show MAML and CommonMean, respectively, for this problem.

Recently, Balcan et al. [2] study the convex online meta-learning setting and show that both approaches achieve the same average task regret. Here, we consider the offline setting. First, the following Proposition shows that both MAML and CommonMean converge to the same meta-parameter. All proofs are in the appendix.

**Proposition 1.** *Let $\eta_t = 1/t$. Assume that $\gamma < 1/\sigma_{\mathbf{x}}^2$. Both $\boldsymbol{\psi}_t$ in MAML (with one inner gradient step) and $\boldsymbol{\theta}_t$ in CommonMean converge to $\bar{\mathbf{w}} = \mathbb{E}_\tau \mathbf{w}_\tau^*$.*

The following Proposition shows that $\bar{\mathbf{w}}$ in Proposition 1 is also the best $\boldsymbol{\psi}$ (for MAML) or $\boldsymbol{\theta}$ (for CommonMean) with the smallest population risk for this meta-learning problem.

**Proposition 2.** *Assume that $\gamma < 1/\sigma_{\mathbf{x}}^2$. We have $\bar{\mathbf{w}} = \text{argmin}_{\boldsymbol{\theta}} \mathbb{E}_\tau \mathbb{E}_{S_\tau} \mathbb{E}_{Q_\tau} \sum_{(\mathbf{x},y) \in Q_\tau}(\mathbf{x}^\top \mathbf{w}_\tau^{(\text{prox})} - y)^2 = \text{argmin}_{\boldsymbol{\psi}} \mathbb{E}_\tau \mathbb{E}_{S_\tau} \mathbb{E}_{Q_\tau} \sum_{(\mathbf{x},y) \in Q_\tau}(\mathbf{x}^\top \mathbf{w}_\tau^{(\text{gd})} - y)^2$.*

During meta-testing, we sample a task $\tau' \sim p(\tau)$ with parameter $\mathbf{w}_{\tau'}^*$. Let $\mathbf{X}_{\tau'}$ be the sample matrix from $S_{\tau'}$, and $\mathbf{y}_{\tau'}$ be the corresponding label vector. We assume that $\mathbf{X}_{\tau'}$ is full rank and $n_s < d$. To simplify notations, we drop the subscript $\tau'$ in the following. Let the singular value decomposition of $\mathbf{X}$ be $\mathbf{U}\boldsymbol{\Sigma}\mathbf{V}^\top$ (where $\boldsymbol{\Sigma} = \text{diag}([\nu_1, \ldots, \nu_{n_s}])$), and $\mathbf{V}^\perp$ be $\mathbf{V}$'s orthogonal complement.

As only forward passes are needed, it is common for the base learner in MAML to perform multiple gradient steps [12]. With the convex loss and linear model here, the base learner can obtain a globally optimal solution $\mathbf{w}^{(\text{gd}\infty)}$ directly (which is equivalent to taking infinite gradient steps). As is common in few-shot learning, the number of support samples is much smaller than feature dimensionality. Hence, $\mathbf{w}^{(\text{gd}\infty)}$ is not unique but depends on the learned initialization. Let $\mathbf{w}^{(\text{gd}\infty)}$ be written as $\mathbf{w}^{(\text{gd}\infty)} = \mathbf{V}\mathbf{a}^{(\text{gd}\infty)} + \mathbf{V}^\perp \mathbf{b}^{(\text{gd}\infty)}$. For gradient descent, its update direction $\mathbf{X}^\top(\mathbf{X}\mathbf{w} - \mathbf{y}) = \mathbf{V}\boldsymbol{\Sigma}\mathbf{U}^\top(\mathbf{X}\mathbf{w} - \mathbf{y})$ is always in the span of $\mathbf{V}$ and so $\mathbf{b}^{(\text{gd}\infty)}$ remains unchanged.

Let $\mathbf{w}^*$ and $\boldsymbol{\theta}$ be written as $\mathbf{w}^* = \mathbf{V}\mathbf{a}^* + \mathbf{V}^\perp \mathbf{b}^*$ and $\boldsymbol{\theta} = \mathbf{V}\mathbf{a}_0 + \mathbf{V}^\perp \mathbf{b}_0$. Moreover, let $\tilde{\mathbf{a}} = \mathbf{a}_0 - \mathbf{a}^*$ and $\tilde{\mathbf{b}} = \mathbf{b}_0 - \mathbf{b}^*$.

**Proposition 3** ([17]). *Assume that $\gamma < \min_{1 \leq j \leq n_s} 1/\nu_j^2$. We have $\mathbb{E}_{\boldsymbol{\xi}}\|\mathbf{w}^{(gd\infty)} - \mathbf{w}^*\|^2 = \|\mathbf{b}^{(gd\infty)} - \mathbf{b}^*\|^2 + \sum_{j=1}^{n_s}\left(\frac{\sigma_\xi}{\nu_j}\right)^2$, where the expectation is over the label noise vector $\boldsymbol{\xi}$.*

For CommonMean, using the Woodbury matrix identity, $\mathbf{w}^{(\text{prox})} = \boldsymbol{\theta} + \mathbf{X}^\top(\lambda\mathbf{I} + \mathbf{X}\mathbf{X}^\top)^{-1}(\mathbf{y} - \mathbf{X}\boldsymbol{\theta}) = \mathbf{V}(\mathbf{a}_0 + \boldsymbol{\Sigma}\mathbf{U}^\top(\lambda\mathbf{I} + \mathbf{X}\mathbf{X}^\top)^{-1}(\mathbf{y} - \mathbf{X}\boldsymbol{\theta})) + \mathbf{V}^\perp\mathbf{b}_0 \equiv \mathbf{V}\mathbf{a}^{(\text{prox})} + \mathbf{V}^\perp\mathbf{b}^{(\text{prox})}$, where $\mathbf{b}^{(\text{prox})} = \mathbf{b}_0$. Assume that $\mathbf{w}^{(\text{gd}\infty)}$ is initialized with $\boldsymbol{\theta}$. Since $\mathbf{b}^{(\text{gd}\infty)}$ remains unchanged, $\mathbf{w}^{(\text{prox})}$ and $\mathbf{w}^{(\text{gd}\infty)}$ only differ in the components lying in the column space of $\mathbf{V}$.

**Proposition 4.** $\mathbb{E}_{\boldsymbol{\xi}}\|\mathbf{w}^{(\text{prox})} - \mathbf{w}^*\|^2 = \|\tilde{\mathbf{b}}\|^2 + \sum_{j=1}^{n_s}\left(\frac{\lambda\tilde{\mathbf{a}}_j}{\lambda+\nu_j^2}\right)^2 + \sum_{j=1}^{n_s}\left(\frac{\sigma_\xi}{(\lambda/\nu_j)+\nu_j}\right)^2$, *where the expectation is over the label noise vector $\boldsymbol{\xi}$.*

As can be seen, when the labels are noise-free ($\sigma_\xi^2 = 0$), $\mathbf{w}^{(\text{gd}\infty)}$ performs better than $\mathbf{w}^{(\text{prox})}$. However, when the labels are noisy, as $n_s < d$, gradient descent always converges to zero training error and overfits the noisy labels. On the other hand, the estimation error of $\mathbf{w}^{(\text{prox})}$ equals to that of $\mathbf{w}^{(\text{gd}\infty)}$ when $\lambda = 0$. For $\lambda > 0$, it trades off between fitting the noisy labels (the last term in Proposition 4) and introducing an estimation bias of $\mathbf{a}^*$ (the second term in Proposition 4).

### 3.2 Proposed Algorithm

It is straightforward to use the dual formulation for the CommonMean algorithm. When the square loss is used as $\ell(\cdot, \cdot)$, it is easy to see that the dual variable has the closed-form solution

$$\boldsymbol{\alpha}_\tau = (\mathbf{I} + \lambda^{-1}\mathbf{X}_\tau\mathbf{X}_\tau^\top)^{-1}(\mathbf{y}_\tau - \mathbf{X}_\tau\boldsymbol{\theta}). \tag{3}$$

Compared with the primal formulation, we only need to invert a $n_s \times n_s$ matrix (instead of the $d \times d$ matrix $\lambda\mathbf{I} + \mathbf{X}_\tau^\top\mathbf{X}_\tau$). In meta-learning, usually $n_s \ll d$ (e.g., $n_s = 5$). From the dual solution $\boldsymbol{\alpha}_\tau$, the primal solution can be recovered as $\mathbf{w}_\tau = \boldsymbol{\theta} + \lambda^{-1}\mathbf{X}_\tau^\top\boldsymbol{\alpha}_\tau$. Given a query example $(\mathbf{x}, y) \in Q_\tau$, the model predicts $\hat{y} = \mathbf{x}^\top\mathbf{w}_\tau = \mathbf{x}^\top\boldsymbol{\theta} + \lambda^{-1}\mathbf{x}^\top\mathbf{X}_\tau^\top\boldsymbol{\alpha}_\tau$. The loss gradient is $\nabla_{\boldsymbol{\theta}}\ell(\hat{y}, y) = \nabla_1\ell(\hat{y}, y)\nabla_{\boldsymbol{\theta}}\hat{y}$, where $\nabla_1\ell(\hat{y}, y)$ denotes the gradient w.r.t. the first argument, and $\nabla_{\boldsymbol{\theta}}\hat{y} = \mathbf{x} + \lambda^{-1}(\nabla_{\boldsymbol{\theta}}\boldsymbol{\alpha}_\tau)^\top\mathbf{X}_\tau\mathbf{x}, \nabla_{\boldsymbol{\theta}}\boldsymbol{\alpha}_\tau = -(\mathbf{I} + \lambda^{-1}\mathbf{X}_\tau\mathbf{X}_\tau^\top)^{-1}\mathbf{X}_\tau$. The complexity of computing $\nabla_{\boldsymbol{\theta}}\ell(\hat{y}, y)$ is thus very low ($\mathcal{O}(n_s^3 + n_s^2 d)$).

The dual formulation also allows introduction of nonlinearity with the kernel trick. Based on deep kernels [42], recent state-of-the-arts (R2D2 [4], MetaOptNet [22], and DKT [26]) propose to use a base kernel in the base learner and update the deep network in the meta-learner. However, their regularizers are not learnable. We consider to learn a proximal regularizer. An input $\mathbf{x}$ is mapped to $\mathbf{z} = \text{NN}(\mathbf{x}; \boldsymbol{\phi})$ in an embedding space $\mathcal{E}$ via a deep network parameterized by $\boldsymbol{\phi}$. With the dual formulation, $\boldsymbol{\theta}$ in (2) allows extra flexibility over [22, 4, 7]. Specifically, $\boldsymbol{\theta}$ becomes a function $f_{\boldsymbol{\theta}} \in \mathcal{H}$, the reproducing kernel Hilbert space (RKHS) corresponding to a given kernel function $\mathcal{K}$ on $\mathcal{E} \times \mathcal{E}$. The base learner obtains a model $f_\tau \in \mathcal{H}$ by minimizing

$$\min_{f \in \mathcal{H}}\sum_{(\mathbf{x}_i, y_i) \in S_\tau}\ell(f(\mathbf{z}_i), y_i) + \frac{\lambda}{2}\|f - f_{\boldsymbol{\theta}}\|_{\mathcal{H}}^2, \tag{4}$$

where $\|\cdot\|_{\mathcal{H}}$ is the norm on $\mathcal{H}$. By setting $f_{\boldsymbol{\theta}} = 0$, this recovers the state-of-the-arts of MetaOpt-Net [22], R2D2 [4], and DKT [26]. However, Proposition 4 suggests that a good $f_{\boldsymbol{\theta}}$ provides good meta-knowledge by biasing the task model.

By the representer theorem [34], the solution of (4) is $f_\tau = f_{\boldsymbol{\theta}} + \sum_{(\mathbf{x}_i, y_i) \in S_\tau}\alpha_{\tau,i}\mathcal{K}_{\mathbf{z}_i}$, where $\mathcal{K}_{\mathbf{z}_i} = \mathcal{K}(\mathbf{z}_i, \cdot) \in \mathcal{H}$, and $\boldsymbol{\alpha}_\tau = [\alpha_{\tau,1}; \ldots; \alpha_{\tau,n_s}]$ is obtained from the convex program

$$\min_{\boldsymbol{\alpha}_\tau}\sum_{(\mathbf{x}_i, y_i) \in S_\tau}\ell(f_\tau(\mathbf{z}_i), y_i) + \boldsymbol{\alpha}_\tau^\top\mathcal{K}(\mathbf{Z}_\tau, \mathbf{Z}_\tau)\boldsymbol{\alpha}_\tau, \tag{5}$$

where $\mathbf{Z}_\tau = [\mathbf{z}_1^\top; \ldots; \mathbf{z}_{n_s}^\top]$, and $\mathcal{K}(\mathbf{Z}_\tau, \mathbf{Z}_\tau)$ is the kernel matrix. Note that the hyperparameter $\lambda$ in (4) is absorbed into $\mathbf{z}$ as the network is learnable. With the square loss, the dual solution of (5) is $\boldsymbol{\alpha}_\tau = (\mathbf{I} + \mathcal{K}(\mathbf{Z}_\tau, \mathbf{Z}_\tau))^{-1}(\mathbf{y}_\tau - f_{\boldsymbol{\theta}}(\mathbf{Z}_\tau))$, where $f_{\boldsymbol{\theta}}(\mathbf{Z}_\tau) = [f_{\boldsymbol{\theta}}(\mathbf{z}_1); \ldots; f_{\boldsymbol{\theta}}(\mathbf{z}_{n_s})]$. For general loss functions, the dual problem has no closed-form solution, but this has only $n_s$ variables (which is usually small) and can be solved efficiently.

After the base learner has obtained the dual solution $\boldsymbol{\alpha}_\tau$, the meta-learner updates $f_{\boldsymbol{\theta}}$ and network parameter $\boldsymbol{\phi}$ by one gradient descent step on the validation loss $\sum_{(\mathbf{x}, y) \in Q_\tau}\ell(\hat{y}, y)$, where $\hat{y} \equiv f_\tau(\mathbf{z}) = f_{\boldsymbol{\theta}}(\mathbf{z}) + \mathcal{K}(\mathbf{Z}_\tau, \mathbf{z})^\top\boldsymbol{\alpha}_\tau$. Using the chain rule, $\nabla_{(\boldsymbol{\theta}, \boldsymbol{\phi})}\ell(\hat{y}, y) = \nabla_1\ell(\hat{y}, y)\nabla_{(\boldsymbol{\theta}, \boldsymbol{\phi})}\hat{y}$. The first component $\nabla_1\ell(\hat{y}, y)$ can be computed directly and the second component is

$$\nabla_{(\boldsymbol{\theta}, \boldsymbol{\phi})}\hat{y} = \nabla_{(\boldsymbol{\theta}, \boldsymbol{\phi})}f_{\boldsymbol{\theta}}(\mathbf{z}) + (\nabla_{(\boldsymbol{\theta}, \boldsymbol{\phi})}\mathcal{K}(\mathbf{Z}_\tau, \mathbf{z}))^\top\boldsymbol{\alpha}_\tau + (\nabla_{(\boldsymbol{\theta}, \boldsymbol{\phi})}\boldsymbol{\alpha}_\tau)^\top\mathcal{K}(\mathbf{Z}_\tau, \mathbf{z}). \tag{6}$$

Both $\nabla_{(\boldsymbol{\theta},\boldsymbol{\phi})} f_{\boldsymbol{\theta}}(\mathbf{z})$ and $\nabla_{(\boldsymbol{\theta},\boldsymbol{\phi})}\mathcal{K}(\mathbf{Z}_\tau, \mathbf{z})$ can be obtained by direct differentiation. By the chain rule, $\nabla_{(\boldsymbol{\theta},\boldsymbol{\phi})}\boldsymbol{\alpha}_\tau = \nabla_{\mathbf{p}}\boldsymbol{\alpha}_\tau \nabla_{(\boldsymbol{\theta},\boldsymbol{\phi})}\mathbf{p}$, where $\mathbf{p} = [f_{\boldsymbol{\theta}}(\mathbf{z}_1); \ldots; f_{\boldsymbol{\theta}}(\mathbf{z}_{n_s}); \mathcal{K}(\mathbf{Z}_\tau, \mathbf{z}_1); \ldots; \mathcal{K}(\mathbf{Z}_\tau, \mathbf{z}_{n_s})] \in \mathbb{R}^{n_s + n_s^2}$ is the input to the dual problem. $\nabla_{(\boldsymbol{\theta},\boldsymbol{\phi})}\mathbf{p}$ can be directly computed. When the square loss is used, $\nabla_{\mathbf{p}}\boldsymbol{\alpha}_\tau = -(\mathbf{I} + \mathcal{K}(\mathbf{Z}_\tau, \mathbf{Z}_\tau))^{-1}[\mathbf{I} \mid \mathbf{I} \otimes \boldsymbol{\alpha}_\tau^\top]$. For a general loss, $\boldsymbol{\alpha}_\tau$ is obtained by solving the convex program. Hence, $\boldsymbol{\alpha}_\tau$ depends implicitly on $\mathbf{p}$ and $\nabla_{\mathbf{p}}\boldsymbol{\alpha}_\tau$ can be obtained by implicit differentiation. Denote the dual objective in (5) by $g(\mathbf{p}, \boldsymbol{\alpha})$. By the implicit function theorem [32], $\nabla_{\mathbf{p}}\boldsymbol{\alpha}_\tau = -\left(\nabla_{\boldsymbol{\alpha}}^2 g(\mathbf{p}, \boldsymbol{\alpha}_\tau)\right)^{-1}\frac{\partial^2}{\partial\mathbf{p}\partial\boldsymbol{\alpha}}g(\mathbf{p}, \boldsymbol{\alpha}_\tau)$, where $\nabla_{\boldsymbol{\alpha}}^2 g(\mathbf{p}, \boldsymbol{\alpha}_\tau) = \sum_{(\mathbf{x}_i,y_i)\in S_\tau}\nabla_1^2\ell(f_\tau(\mathbf{z}_i), y_i)\mathcal{K}(\mathbf{Z}_\tau, \mathbf{z}_i)\mathcal{K}(\mathbf{Z}_\tau, \mathbf{z}_i)^\top + \mathcal{K}(\mathbf{Z}_\tau, \mathbf{Z}_\tau)$, $\frac{\partial^2}{\partial\mathbf{p}\partial\boldsymbol{\alpha}}g(\mathbf{p}, \boldsymbol{\alpha}_\tau) = [\mathcal{K}(\mathbf{Z}_\tau, \mathbf{Z}_\tau)\mathbf{D} \mid (\mathcal{K}(\mathbf{Z}_\tau, \mathbf{Z}_\tau)\mathbf{D}) \otimes \boldsymbol{\alpha}_\tau^\top + \mathbf{v}^\top \otimes \mathbf{I} + \mathbf{I} \otimes \boldsymbol{\alpha}_\tau^\top]$, $\mathbf{D} = \mathrm{diag}([\nabla_1^2\ell(f_\tau(\mathbf{z}_1), y_1); \ldots; \nabla_1^2\ell(f_\tau(\mathbf{z}_{n_s}), y_{n_s})])$, $\mathbf{v} = [\nabla_1\ell(f_\tau(\mathbf{z}_1), y_1); \ldots; \nabla_1\ell(f_\tau(\mathbf{z}_{n_s}), y_{n_s})]$, where $\otimes$ is the Kronecker product. The whole procedure, called MetaProx, is shown in Algorithm 3. Let $n_{\boldsymbol{\phi}}$ and $n_{\boldsymbol{\theta}}$ be the numbers of parameters in $\boldsymbol{\phi}$ and $\boldsymbol{\theta}$, respectively. Computing $\nabla_{(\boldsymbol{\theta},\boldsymbol{\phi})}\ell(\hat{y}, y)$ takes $\mathcal{O}(n_s^3 + n_s^2(n_{\boldsymbol{\theta}} + n_{\boldsymbol{\phi}}))$ time, which is linear in the number of meta-parameters. This is lower than the other meta-learning algorithms (e.g., MAML [12] with single step takes $\mathcal{O}(n_{\boldsymbol{\phi}}^2)$ time, iMAML [28]: $\mathcal{O}(n_{\boldsymbol{\phi}}^3)$, CommonMean [7]: $\mathcal{O}(d^3)$).

---

**Algorithm 3** MetaProx.

---

**Require:** step size $\eta_t$, batch size $b$;
1: **for** $t = 1, 2, \cdots, T$ **do**
2:     sample a batch $\mathcal{B}_t$ of tasks from $\mathcal{T}$;
3:     base learner:
4:     **for** $\tau \in \mathcal{B}_t$ **do**
5:         $\mathbf{z}_i = \mathrm{NN}(\mathbf{x}_i; \boldsymbol{\phi}_t)$ for each $(\mathbf{x}_i, y_i) \in S_\tau$;
6:         $f_\tau(\mathbf{z}; \boldsymbol{\alpha}) \equiv f_{\boldsymbol{\theta}_t}(\mathbf{z}) + \mathcal{K}(\mathbf{Z}_\tau, \mathbf{z})^\top \boldsymbol{\alpha}$ denote the task model w.r.t. dual variables;
7:         $\boldsymbol{\alpha}_\tau = \mathrm{argmin}_{\boldsymbol{\alpha}} \sum_{(\mathbf{x}_i,y_i)\in S_\tau}\ell(f_\tau(\mathbf{z}_i; \boldsymbol{\alpha}), y_i) + \boldsymbol{\alpha}^\top \mathcal{K}(\mathbf{Z}_\tau, \mathbf{Z}_\tau)\boldsymbol{\alpha}$;
8:         $\mathbf{g}_\tau = \sum_{(\mathbf{x},y)\in Q_\tau}\nabla_{(\boldsymbol{\theta}_t,\boldsymbol{\phi}_t)}\ell(\hat{y}, y)$, where $\hat{y} = f_\tau(\mathbf{z}; \boldsymbol{\alpha}_\tau)$ and $\mathbf{z} = \mathrm{NN}(\mathbf{x}; \boldsymbol{\phi}_t)$;
9:     **end for**
10:    meta-learner: $(\boldsymbol{\theta}_{t+1}, \boldsymbol{\phi}_{t+1}) = (\boldsymbol{\theta}_t, \boldsymbol{\phi}_t) - \frac{\eta_t}{b}\sum_{\tau\in\mathcal{B}_t}\mathbf{g}_\tau$;
11: **end for**

---

**Classification.** We consider extension from regression to $N$-way classification. For task $\tau$, the base learner learns the model $\mathbf{f}_\tau = [f_\tau^{(1)}; \ldots; f_\tau^{(N)}]$ for the $N$ classes by minimizing

$$\min_{f^{(1)}, \ldots, f^{(N)} \in \mathcal{H}} \sum_{(\mathbf{x}_i,y_i)\in S_\tau} \ell(\hat{\mathbf{y}}_i, y_i) + \frac{\lambda}{2}\sum_{c=1}^N \|f^{(c)} - f_{\boldsymbol{\theta}^{(c)}}\|_{\mathcal{H}}^2, \tag{7}$$

where $\hat{\mathbf{y}}_i = [f^{(1)}(\mathbf{z}_i); \ldots; f^{(N)}(\mathbf{z}_i)]$, and $f_{\boldsymbol{\theta}^{(1)}}, \ldots, f_{\boldsymbol{\theta}^{(N)}} \in \mathcal{H}$ are functions learned by the meta-learner. By the representer theorem [34], $\mathbf{f}_\tau = [f_{\boldsymbol{\theta}^{(1)}} + \mathcal{K}(\mathbf{Z}_\tau, \cdot)^\top\boldsymbol{\alpha}_\tau^{(1)}; \ldots; f_{\boldsymbol{\theta}^{(N)}} + \mathcal{K}(\mathbf{Z}_\tau, \cdot)^\top\boldsymbol{\alpha}_\tau^{(N)}]$, where $[\boldsymbol{\alpha}_\tau^{(1)}; \cdots; \boldsymbol{\alpha}_\tau^{(N)}]$ is obtained from the convex program $\min_{\boldsymbol{\alpha}^{(1)}, \ldots, \boldsymbol{\alpha}^{(N)}}\sum_{(\mathbf{x}_i,y_i)\in S_\tau}\ell(\hat{\mathbf{y}}_i, y_i) + \sum_{c=1}^N \boldsymbol{\alpha}^{(c)\top}\mathcal{K}(\mathbf{Z}_\tau, \mathbf{Z}_\tau)\boldsymbol{\alpha}^{(c)}$. As $[\boldsymbol{\alpha}_\tau^{(1)}; \cdots; \boldsymbol{\alpha}_\tau^{(N)}] \in \mathbb{R}^{Nn_s}$ and both $N, n_s$ are typically very small (e.g., $N = 5, n_s = 5$ in 5-way 5-shot classification), this convex program can be solved efficiently. The meta-learner then updates the network parameter $\boldsymbol{\phi}$ and $\{f_{\boldsymbol{\theta}^{(1)}}, \ldots, f_{\boldsymbol{\theta}^{(N)}}\}$ by one gradient descent step on the validation loss $\sum_{(\mathbf{x}_i,y_i)\in Q_\tau}\ell(\hat{\mathbf{y}}_i, y_i)$. Computing $\nabla_{(\boldsymbol{\theta}^{(1)}, \ldots, \boldsymbol{\theta}^{(N)}, \boldsymbol{\phi})}\ell(\hat{\mathbf{y}}, y)$ takes $\mathcal{O}((Nn_s)^3 + (Nn_s)^2(n_{\boldsymbol{\phi}} + \sum_{c=1}^N n_{\boldsymbol{\theta}^{(c)}}))$ time, where $n_{\boldsymbol{\theta}^{(c)}}$ is the size of $\boldsymbol{\theta}^{(c)}$. This is again linear in the number of meta-parameters and thus very efficient.

### 3.3 Theoretical Analysis

Let $\mathcal{L}_{\mathrm{meta}}(\boldsymbol{\theta}, \boldsymbol{\phi}) = \frac{1}{|\mathcal{T}|}\sum_{\tau\in\mathcal{T}}\sum_{(\mathbf{x},y)\in Q_\tau}\ell(f_{\boldsymbol{\theta}}(\mathbf{z}; \boldsymbol{\alpha}_\tau), y)$ be the empirical loss of $\mathbb{E}_{\tau\sim p(\tau)}\sum_{(\mathbf{x},y)\in Q_\tau}\ell(f_{\boldsymbol{\theta}}(\mathbf{z}; \boldsymbol{\alpha}_\tau), y)$, where $\mathbf{z} = \mathrm{NN}(\mathbf{x}; \boldsymbol{\phi})$. With the linear kernel and square loss, the dual solution (3) is affine in the meta-parameter, and so is the primal solution $\mathbf{w}_\tau = \boldsymbol{\theta} + \lambda^{-1}\mathbf{X}_\tau^\top\boldsymbol{\alpha}_\tau$. Thus, the meta-loss $\mathcal{L}_{\mathrm{meta}}(\boldsymbol{\theta}, \boldsymbol{\phi})$ is convex and convergence follows from convex optimization [5, 7]. After introducing nonlinearity, the meta-loss is no longer convex. The following introduces Lipschitz-smoothness assumptions, which have been commonly used in stochastic non-convex optimization [15, 31] and meta-learning in non-convex settings [11, 43].

**Assumption 1** (Smoothness). *(i) The deep network* $\mathrm{NN}(\mathbf{x}; \boldsymbol{\phi})$ *is Lipschitz-smooth, i.e.,* $\|\nabla_{\boldsymbol{\phi}} \mathrm{NN}(\mathbf{x}; \boldsymbol{\phi}) - \nabla_{\boldsymbol{\phi}} \mathrm{NN}(\mathbf{x}; \boldsymbol{\phi}')\| \leq \beta_1 \|\boldsymbol{\phi} - \boldsymbol{\phi}'\|$ *with a Lipschitz constant* $\beta_1 > 0$; *(ii) the kernel* $\mathcal{K}(\mathbf{z}, \mathbf{z}')$ *is Lipschitz-smooth w.r.t.* $(\mathbf{z}, \mathbf{z}')$; *(iii)* $f_{\boldsymbol{\theta}}(\mathbf{z})$ *is Lipschitz-smooth w.r.t.* $(\boldsymbol{\theta}, \mathbf{z})$; *(iv)* $\nabla_1^2 \ell(\hat{y}, y)$ *is Lipschitz w.r.t.* $\hat{y}$, *i.e.,* $|\nabla_1^2 \ell(\hat{y}, y) - \nabla_1^2 \ell(\hat{y}', y)| \leq \beta_2 |\hat{y} - \hat{y}'|$ *with a Lipschitz constant* $\beta_2$; *(v)* $\mathbb{E}_{\tau \sim \mathcal{T}} \|\nabla_{(\boldsymbol{\theta}, \boldsymbol{\phi})} \sum_{(\mathbf{x}, y) \in Q_\tau} \ell(f_{\boldsymbol{\theta}}(\mathbf{z}; \boldsymbol{\alpha}_\tau), y) - \nabla_{(\boldsymbol{\theta}, \boldsymbol{\phi})} \mathcal{L}_{meta}(\boldsymbol{\theta}, \boldsymbol{\phi})\|^2 = \sigma_{\mathbf{g}}^2$, *where* $\tau \sim \mathcal{T}$ *denotes uniformly sample a task from* $\mathcal{T}$.

The following Lemma guarantees smoothness of the meta-loss.

**Lemma 1.** $\mathcal{L}_{meta}(\boldsymbol{\theta}, \boldsymbol{\phi})$ *is Lipschitz-smooth w.r.t.* $(\boldsymbol{\theta}, \boldsymbol{\phi})$ *with a Lipschitz constant* $\beta_{meta}$.

**Theorem 1.** *Let the step size be* $\eta_t = \min(1/\sqrt{T}, 1/2\beta_{meta})$. *Algorithm 3 satisfies* $\min_{1 \leq t \leq T} \mathbb{E} \|\nabla_{(\boldsymbol{\theta}_t, \boldsymbol{\phi}_t)} \mathcal{L}_{meta}(\boldsymbol{\theta}_t, \boldsymbol{\phi}_t)\|^2 = \mathcal{O}\left(\sigma_{\mathbf{g}}^2 / \sqrt{T}\right)$, *where the expectation is taken over the random training samples.*

This rate is the same as MAML [11, 19] and Meta-MinibatchProx [43]. For MAML with $J > 1$ gradient steps, Ji et al. [19] assumes that the step size in the inner loop is of the order $1/J$. This slows down inner loop learning when $J$ is large. On the other hand, MetaProx does not have this restriction, as its meta-gradient depends only on the last iterate rather than all iterates along the trajectory.

Next, we study the global convergence of MetaProx. Prior work [13, 43] focus on the case where $\mathcal{L}_{\mathrm{meta}}(\boldsymbol{\theta}, \boldsymbol{\phi})$ is strongly convex in $(\boldsymbol{\theta}, \boldsymbol{\phi})$. This can be restrictive in deep learning. We instead only require $\ell(\hat{y}, y)$ to be strongly convex in $\hat{y}$. This assumption is easily met by commonly-used loss functions such as the square loss and logistic loss with a compact domain. A recent work [40] studies the global convergence of MAML in over-parameterized neural networks. Over-parameterization is closely related to the assumption of uniform conditioning [18, 21, 23].

**Assumption 2** (Uniform conditioning [23]). *A multivariable function* $\mathcal{M}(\boldsymbol{\theta}, \boldsymbol{\phi})$ *is* $\mu$-*uniformly conditioning if its tangent kernel [18] satisfies* $\min_{(\boldsymbol{\theta}, \boldsymbol{\phi})} \lambda_{\min}(\nabla_{(\boldsymbol{\theta}, \boldsymbol{\phi})} \mathcal{M}(\boldsymbol{\theta}, \boldsymbol{\phi}) \nabla_{(\boldsymbol{\theta}, \boldsymbol{\phi})}^\top \mathcal{M}(\boldsymbol{\theta}, \boldsymbol{\phi})) \geq \mu > 0$, *where* $\lambda_{\min}(\cdot)$ *is the smallest eigenvalue of the matrix argument.*

Assume that the loss $\ell(\cdot, \cdot)$ is $\rho$-strongly convex w.r.t. the first argument and Assumption 1 holds. Let $\mathbf{x}_{\tau,j}$ be the $j$th query example of task $\tau$, $\mathbf{z}_{\tau,j}$ be its embedding, and $\hat{y}_{\tau,j} = f_{\boldsymbol{\theta}}(\mathbf{z}_{\tau,j}) + \mathcal{K}(\mathbf{Z}_\tau, \mathbf{z}_{\tau,j})^\top \boldsymbol{\alpha}_\tau$ be its prediction, where $\boldsymbol{\alpha}_\tau$ is the dual solution. Let $\mathcal{M}(\boldsymbol{\theta}, \boldsymbol{\phi}) = \left[\hat{y}_{\tau_1, 1}; \ldots; \hat{y}_{\tau_1, n_q}; \ldots; \hat{y}_{\tau_{|\mathcal{T}|}, 1}; \ldots; \hat{y}_{\tau_{|\mathcal{T}|}, n_q}\right]$ be an auxiliary function which maps the meta-parameter to predictions on all query examples. The following Theorem shows that the proposed algorithm converges to a global minimum of the empirical risk $\mathcal{L}_{\mathrm{meta}}(\boldsymbol{\theta}, \boldsymbol{\phi})$ at the rate of $\mathcal{O}(\sigma_{\mathbf{g}}^2 / \sqrt{T})$. The rate is improved to exponential if the meta-learner adopts full gradient descent.

**Theorem 2.** *Assume that* $\mathcal{M}(\boldsymbol{\theta}, \boldsymbol{\phi})$ *is uniform conditioning. (i) Let* $\eta_t = \min(1/\sqrt{T}, 1/2\beta_{meta})$. *Algorithm 3 satisfies* $\min_{1 \leq t \leq T} \mathbb{E} \mathcal{L}_{meta}(\boldsymbol{\theta}_t, \boldsymbol{\phi}_t) - \min_{(\boldsymbol{\theta}, \boldsymbol{\phi})} \mathcal{L}_{meta}(\boldsymbol{\theta}, \boldsymbol{\phi}) = \mathcal{O}\left(\sigma_{\mathbf{g}}^2 / \sqrt{T}\right)$, *where the expectation is taken over the random training samples. (ii) Let* $\eta_t = \eta < \min(1/2\beta_{meta}, 4|\mathcal{T}|/\rho\mu)$ *and* $\mathcal{B}_t = \mathcal{T}$. *Algorithm 3 satisfies* $\mathcal{L}_{meta}(\boldsymbol{\theta}_t, \boldsymbol{\phi}_t) - \min_{(\boldsymbol{\theta}, \boldsymbol{\phi})} \mathcal{L}_{meta}(\boldsymbol{\theta}, \boldsymbol{\phi}) = \mathcal{O}((1 - \eta\rho\mu/4|\mathcal{T}|)^t)$.

## 4 Experiments

### 4.1 Few-shot Regression

**Data sets**. Experiments are performed on three data sets.

(i) *Sine*. This is the sinusoid regression problem in [12]. Samples $x$'s are uniformly sampled from $[-5, 5]$. Each task $\tau$ learns a sine function $y = a_\tau \sin(x + b_\tau) + \xi$, where $a_\tau \in [0.1, 5]$, $b_\tau \in [0, \pi]$, and $\xi \sim \mathcal{N}(0, \sigma_\xi^2)$ is the label noise. We consider both $\sigma_\xi^2 = 0$ (noise-free) and $\sigma_\xi^2 = 1$. In addition to the 5-shot setting in [12], we also evaluate on the more challenging 2-shot setting. We randomly generate a meta-training set of 8000 tasks, a meta-validation set of 1000 tasks for early stopping, and a meta-testing set of 2000 tasks for performance evaluation.

(ii) *Sale*. This is a real-world dataset from [36], which contains weekly purchased quantities of 811 products over 52 weeks. For each product (task), a sample is to predict the sales quantity for the current week from sales quantities in the previous 5 weeks. Thus, each product contains 47 samples. We evaluate on the 5-shot and 1-shot settings. We randomly split the tasks into a meta-training set of 600 tasks, a meta-validation set of 100 tasks, and a meta-testing set of 111 tasks.

(iii) *QMUL*, which is a multiview face dataset [16] from Queen Mary University of London. This consists of grayscale face images of 37 people (32 for meta-training and 5 for meta-testing). We follow the setting in [26] and evaluate the model on 10-shot regression. Each person has 133 facial images covering a viewsphere of $\pm 90°$ in yaw and $\pm 30°$ in tilt at $10°$ increment. A task is a trajectory taken from the discrete manifold for images from the same person. The regression goal is to predict the tilt given an image. In the *in-range* setting, meta-training tasks are sampled from the entire manifold. In the more challenging *out-of-range* setting, meta-training tasks are sampled from the sub-manifold with yaw in $[-90°, 0°]$. In both settings, meta-testing tasks are sampled from the entire manifold. We randomly sample 2400 tasks for meta-training, and 500 tasks for meta-testing. As in [26], we do not use a meta-validation set since the dataset is small.

**Network Architecture**. For *Sine* and *Sale*, we use the network in [12], which is a small multilayer perceptron with two hidden layers of size 40 and ReLU activation. For *QMUL*, we use the three-layer convolutional neural network in [26]. The embeddings are always from the last hidden layer. We use a simple linear kernel as base kernel, and $f_{\boldsymbol{\theta}}(\mathbf{z}) = \boldsymbol{\theta}^{\top}\mathbf{z}$.

**Implementation Details**. We use the Adam optimizer [20] with a learning rate of $0.001$. Each mini-batch has 16 tasks. For *Sine* and *Sale*, the model ($\phi$ and $f_{\boldsymbol{\theta}}$) is meta-trained for $40,000$ iterations. To prevent overfitting on the meta-training set, we evaluate the meta-validation performance every 500 iterations, and stop training when the loss on the meta-validation set has no significant improvement for 10 consecutive evaluations. For *QMUL*, we follow [26] and meta-train the model for 100 iterations. We repeat each experiment 30 times. For performance evaluation, we use the average mean squared error (MSE) on the meta-testing set.

**Baselines**. On *Sine* and *Sale*, we compare MetaProx with CommonMean [7], MAML [12], MetaOptNet-RR [22], Meta-MinibatchProx [43], and iMAML [28]. CommonMean is a linear model, and MetaOptNet-RR is equivalent to MetaProx when $f_{\boldsymbol{\theta}} = 0$. Following [12], we set the number of inner gradient steps for MAML to 1 during meta-training, and 20 during meta-validation and meta-testing. Both Meta-MinibatchProx [43] and iMAML [28] are meta-regularization approaches. For *QMUL*, we compare MetaProx with the baselines reported in [26] (namely, DKT [26], Feature Transfer [10], and MAML). As further baselines, we also compare with Meta-MinibatchProx and MetaOptNet-RR to evaluate the improvement of MetaProx due to the learnable $f_{\boldsymbol{\theta}}$.

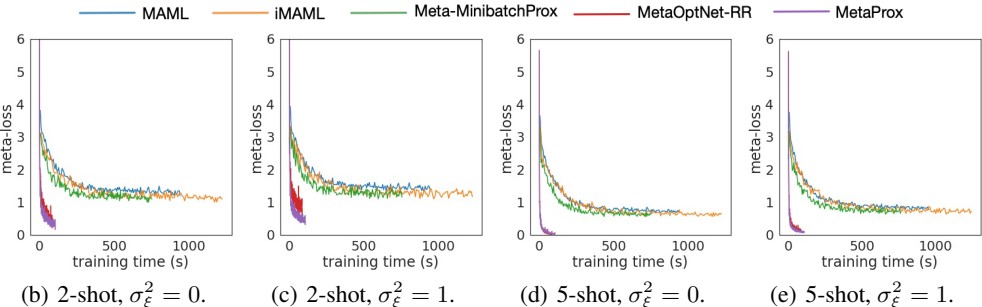

(b) 2-shot, $\sigma_{\xi}^2 = 0$.    (c) 2-shot, $\sigma_{\xi}^2 = 1$.    (d) 5-shot, $\sigma_{\xi}^2 = 0$.    (e) 5-shot, $\sigma_{\xi}^2 = 1$.

Figure 1: Convergence curves for few-shot sinusoid regression. Best viewed in color.

**Results on Sine**. Figure 1 shows the convergence curves of MetaProx and the baselines. We do not show the convergence of CommonMean, as it does not use a neural network backbone as the other methods. As can be seen, MetaProx converges much faster and better than the non-kernel-based methods (MAML, iMAML and Meta-MinibatchProx). In the 2-shot settings, MetaProx converges to a loss smaller than that of MetaOptNet-RR.

Figure 2 shows the learned functions on 2 meta-testing tasks ($\tau_1$ with ($a = 4.6$, $b = 3.2$) and $\tau_2$ with ($a = 3.7$, $b = 0.5$)) in the 5-shot setting and more challenging 2-shot setting. As can be seen, MetaProx always fits the target curve well. Though MAML, iMAML and Meta-MinibatchProx can fit the support samples, it performs worse in regions far from the support samples. This is especially noticeable in the 2-shot setting.

Table 1 shows the MSE on the meta-testing set. Obviously, CommonMean (a linear model) fails in this nonlinear regression task. MetaProx and MetaOptNet-RR significantly outperform the other baselines. MetaProx (with the learned $f_{\boldsymbol{\theta}}$) performs better than MetaOptNet-RR, particularly when the data is noisy.

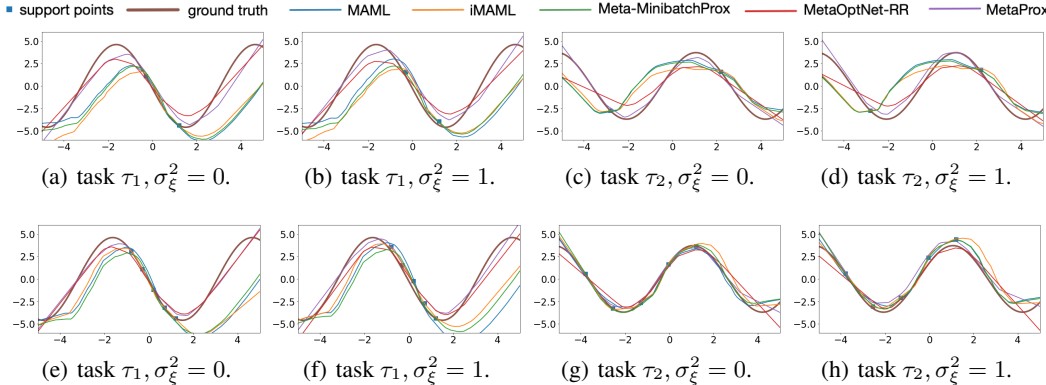

Figure 2: Sinusoid regression: Two meta-testing tasks $\tau_1$ and $\tau_2$ with different $\sigma_\xi$'s in 2-shot ((a)–(d)) and 5-shot ((e)–(h)) settings. Best viewed in color.

Table 1: Average MSE (with $95\%$ confidence intervals) of few-shot regression on the *Sine* and *Sale* datasets. (The confidence intervals in *Sale* experiments are $\pm 0.001$ for all methods)

|  | *Sine* (2-shot) | | *Sine* (5-shot) | | *Sale* | |
|---|---|---|---|---|---|---|
|  | noise-free | noisy | noise-free | noisy | 1-shot | 5-shot |
| CommonMean [7] | $4.58 \pm 0.07$ | $4.59 \pm 0.07$ | $4.29 \pm 0.06$ | $4.31 \pm 0.06$ | 0.090 | 0.074 |
| MAML [12] | $1.24 \pm 0.12$ | $1.91 \pm 0.13$ | $0.41 \pm 0.03$ | $1.15 \pm 0.05$ | 0.069 | 0.063 |
| iMAML [28] | $1.12 \pm 0.11$ | $1.84 \pm 0.10$ | $0.38 \pm 0.02$ | $1.02 \pm 0.05$ | 0.068 | 0.063 |
| Meta-MinibatchProx [43] | $1.15 \pm 0.08$ | $1.87 \pm 0.09$ | $0.37 \pm 0.02$ | $1.01 \pm 0.03$ | 0.081 | 0.064 |
| MetaOptNet-RR [22] | $0.18 \pm 0.01$ | $0.79 \pm 0.01$ | $0.01 \pm 0.00$ | $0.19 \pm 0.01$ | 0.088 | 0.068 |
| MetaProx (proposed) | $\mathbf{0.11 \pm 0.01}$ | $\mathbf{0.43 \pm 0.01}$ | $\mathbf{0.01 \pm 0.00}$ | $\mathbf{0.13 \pm 0.01}$ | **0.061** | **0.060** |

Table 2: Average MSE (with $95\%$ confidence intervals) of few-shot regression on *QMUL* (10-shot). Results of the first four methods are from [26].

| method | in-range | out-of-range |
|---|---|---|
| Feature Transfer [10] | $0.22 \pm 0.03$ | $0.18 \pm 0.01$ |
| MAML [12] | $0.21 \pm 0.01$ | $0.18 \pm 0.02$ |
| DKT + RBF [26] | $0.12 \pm 0.04$ | $0.14 \pm 0.03$ |
| DKT + Spectral [26] | $0.10 \pm 0.02$ | $0.11 \pm 0.02$ |
| Meta-MinibatchProx [43] | $0.171 \pm 0.022$ | $0.193 \pm 0.025$ |
| MetaOptNet-RR [22] | $0.021 \pm 0.007$ | $0.039 \pm 0.009$ |
| MetaProx (proposed) | $\mathbf{0.012 \pm 0.003}$ | $\mathbf{0.020 \pm 0.005}$ |

**Results on Sale**. As can be seen from Table 1, the linear model (CommonMean) performs poorly as expected. MetaProx again outperforms the other baselines, particularly in the more challenging 1-shot setting.

**Results on QMUL**. Table 2 shows that MetaProx achieves the lowest MSE and the kernel methods (DKT+RBF, DKT+Spectral, MetaOptNet-RR, and MetaProx) perform better than non-kernel-based methods (Feature Transfer, MAML, and Meta-MinibatchProx). MetaProx with the learnable $f_{\boldsymbol{\theta}}$ reduces the errors of MetaOptNet-RR by half.

### 4.2 Few-shot Classification

**Datasets.** We use the standard 5-way $K$-shot setting ($K = 1$ or $5$) on the *mini-ImageNet* [39] dataset, which consists of 100 randomly chosen classes from *ILSVRC*-2012 [33]. Each class contains $600 \, 84 \times 84$ images. We use the commonly-used split in [30]: the 100 classes are randomly split into 64 for meta-training, 16 for meta-validation, and 20 for meta-testing.

**Network Architecture.** For the network backbone, we use the *Conv4* in [12, 39] and *ResNet-12* in [22]. As the cosine similarity is more effective than $\ell_2$ distance in few-shot classification [6], we adopt the cosine kernel $\mathcal{K}(\mathbf{z}, \mathbf{z}') = \cos(\mathbf{z}, \mathbf{z}')$ as base kernel, where $\mathbf{z}$ is the embedding of sample $\mathbf{x}$ extracted from the last hidden layer as in regression. For each $c = 1, \ldots, 5$, $f_{\boldsymbol{\theta}^{(c)}} = [\mathcal{K}_{\mathbf{q}^{(1)}}; \ldots; \mathcal{K}_{\mathbf{q}^{(5)}}]^\top \boldsymbol{\theta}^{(c)}$ is a weighted prototype classifier on the embedding space, where

Table 3: Accuracies (with $95\%$ confidence intervals) of 5-way few-shot classification on *mini-ImageNet* using *Conv4*. [†] means that the result is obtained by rerunning the code with our setup here. Other results from their original publications (Result on the 5-shot setting is not reported in iMAML [28]).

| method | 1-shot | 5-shot |
| --- | --- | --- |
| MAML [12] | $48.7 \pm 1.8$ | $63.1 \pm 0.9$ |
| FOMAML [12] | $48.1 \pm 1.8$ | $63.2 \pm 0.9$ |
| REPTILE [25] | $50.0 \pm 0.3$ | $66.0 \pm 0.6$ |
| iMAML [28] | $49.0 \pm 1.8$ | $-$ |
| Meta-MinibatchProx [43] | $50.8 \pm 0.9$ | $67.4 \pm 0.9$ |
| ANIL [27] | $46.7 \pm 0.4$ | $61.5 \pm 0.5$ |
| R2D2 [4] | $49.5 \pm 0.2$ | $65.4 \pm 0.3$ |
| ProtoNet [35] | $49.4 \pm 0.8$ | $68.2 \pm 0.7$ |
| MetaOptNet-SVM(lin)[†] [22] | $49.8 \pm 0.9$ | $66.9 \pm 0.7$ |
| MetaOptNet-SVM(cos)[†] [22] | $50.1 \pm 0.9$ | $67.2 \pm 0.6$ |
| MetaProx (proposed) | $\mathbf{52.4 \pm 1.0}$ | $\mathbf{68.8 \pm 0.8}$ |

$\mathbf{q}^{(1)}, \ldots, \mathbf{q}^{(5)}$ are the class centroids computed from $S_\tau$, and the weights $\{\boldsymbol{\theta}^{(1)}, \ldots, \boldsymbol{\theta}^{(5)}\}$ are meta-parameters.

**Baselines.** We compare MetaProx with the state-of-the-arts: (i) meta-initialization: MAML [12] and its variants FOMAML [12], and REPTILE [25]; (ii) meta-regularization: iMAML [28] and Meta-MinibatchProx [43]; and (iii) metric learning: ANIL [27], R2D2 [4], ProtoNet [35], and MetaOptNet [22] with SVM using the linear kernel and cosine kernel. .

Table 4: Accuracies (with $95\%$ confidence intervals) of 5-way few-shot classification on *mini-ImageNet* using *ResNet-12*. [†] means that the result is obtained by rerunning the code with our setup here.

| method | 1-shot | 5-shot |
| --- | --- | --- |
| FOMAML[†] [12] | $57.41 \pm 0.71$ | $72.12 \pm 0.54$ |
| ANIL[†] [27] | $59.66 \pm 0.68$ | $73.28 \pm 0.49$ |
| ProtoNet [35] | $59.25 \pm 0.64$ | $75.60 \pm 0.48$ |
| MetaOptNet-SVM(lin)[†] [22] | $62.31 \pm 0.64$ | $78.21 \pm 0.42$ |
| MetaOptNet-SVM(cos)[†] [22] | $62.75 \pm 0.42$ | $78.68 \pm 0.24$ |
| MetaProx (proposed) | $\mathbf{63.82 \pm 0.23}$ | $\mathbf{79.12 \pm 0.18}$ |

**Implementation Details.** The entire model is trained end-to-end. $\ell(\hat{\mathbf{y}}, y)$ is the cross-entropy loss. The CVXPYLayers package [1] is used to solve the dual problem. We train the model for $80,000$ iterations, and each mini-batch has 4 tasks. We use the Adam optimizer [20] with an initial learning rate of $0.001$, which is then reduced by half every $2,500$ iterations. To prevent overfitting, we evaluate the meta-validation performance every 500 iterations, and stop training when the meta-validation accuracy has no significant improvement for 10 consecutive evaluations. We report the classification accuracy averaged over 600 tasks randomly sampled from the meta-testing set.

**Results.** Tables 3 and 4 show the results for *Conv4* and *ResNet-12*, respectively. As can be seen, MetaProx is always the best. Compared with MetaOptNet-SVM, MetaProx is better due to the learnable regularizer.

## 5  Conclusion

In this paper, we proposed MetaProx, an effective meta-regularization algorithm for meta-learning. MetaProx combines deep kernel and meta-regularization. By reformulating the problem in the dual space, a learnable proximal regularizer is introduced to the base learner. The meta-parameters in the regularizer and network are updated by the meta-learner. We also established convergence of MetaProx. Extensive experiments on standard datasets for regression and classification verify the effectiveness of learning a proximal regularizer.

## Acknowledgements

This work is supported by NSFC grant 62076118.

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
