# OpenReview forum: "Effective Meta-Regularization by Kernelized Proximal Regularization"
_NeurIPS.cc/2021/Conference — NeurIPS 2021 Poster_

### Official Review · Reviewer_Nvyf · 2021-07-09

**Rating:** 6
**Confidence:** 3

**Summary:**

The authors proposed an effective meta-regularization algorithm (MetaProx) for meta-learning. A learnable proximal regularizer with deep kernel is introduced to learn the base learner. The computation complexity of meta-gradient is linear in the number of meta-parameters. Experiments on synthetic dataset and mini-ImageNet show that the proposed method outperforms the popular few-shot learning methods.

**Limitations And Societal Impact:**

The limitation is not provided. No negative societal impact.

**Main Review:**

1.	This paper is well-written. The research problem and proposed method are clearly introduced.
2.	It is good to have theoretical analysis on the smoothness of the meta-loss and the convergence.
3.	The proposed method is verified to be effective in few-shot regression and classification tasks.
4.	The paper states that the new method is much faster than other meta-learning methods e.g. MAML. However, the paper lacks the comparison of training and converging time.
5.	In few-shot classification experiments, the simple Conv4 network is used. However, the state-of-the-art few-shot learning methods use ResNet and Wide Residual Networks as the backbone. Does the proposed method still achieve better performance than the competitors with large models?


**Time Spent Reviewing:**

3

---

> ### Author Response · Authors · 2021-08-10
> **Reply to Reviewer Nvyf**
>
> Thanks for your review and suggestions.
>
> 1. "The paper states that the new method is much faster than other meta-learning methods e.g. MAML. However, the paper lacks the comparison of training and converging time."
>
> - As suggested, we add convergence curves of MetaProx for few-shot sinusoid regression in Section 4.1. We do not show the convergence of CommonMean, as it does not use a neural network backbone as the other methods. As shown in the figures ([https://www.dropbox.com/s/ks9t2ts8lhui3u1/convergence_curves.pdf?dl=0](https://www.dropbox.com/s/ks9t2ts8lhui3u1/convergence_curves.pdf?dl=0)), MetaProx converges much faster than MAML and iMAML. We will add these in the final version.
>
> 2. "In few-shot classification experiments, the simple Conv4 network is used. However, the state-of-the-art few-shot learning methods use ResNet and Wide Residual Networks as the backbone. Does the proposed method still achieve better performance than the competitors with large models?"
>
> - As suggested, we add an experiment using the ResNet-12, please refer to the table in our reply to Reviewer pmQA. As can be seen, MetaProx outperforms the other baselines.

---

> > ### Comment · Reviewer_Nvyf · 2021-08-17
> > **Thanks for the response!**
> >
> > Thank you for the abundant results and explanation. I have no concern after reading it.

---

### Official Review · Reviewer_S5bU · 2021-07-13

**Rating:** 6
**Confidence:** 4

**Summary:**

This paper introduces a new functional regularisation for meta learning. Often times meta learning algorithms can overfit to the training datasets and hence an introduction of a regularisation in terms of the outer loop parameters is a sensible choice. In particular, they build upon the Common mean algorithm and consider the dual form in which they can derive closed form solutions to the optimisation problem. Their main contribution is that the method remains computationally cheap, competitive in terms of performance and provide some theoretical insights into their proposed method MetaProx.

**Limitations And Societal Impact:**

Overall the paper proposes a new simple method for regularising the meta learning training. The write up of the paper could be improved in particular the dual problem setup as well as their proposed computational tricks as mentioned above. Also in terms of the experiments i believe that methods mentioned above should be included for completeness or at least justified why they weren't added. I understand that the proposed method improves upon metaoptnet but that doesn't mean that it is necessarily the best method and hence should be compared to others as well.

I will be awaiting the rebuttal and am happy to raise my score if the above have been addressed.

**Main Review:**

Reason to accept the paper:
- The paper introduces a novel regularisation for meta learning in the function space and obtains closed-form solutions to the ir optimisation problem.
-  The proposed method is computationally efficient and achieves competitive results compared to previous methods.
-  In particular they show that setting f_{\theta} = 0 reverts back to the popular MetaOPTNet algorithm and demonstrate that their additional regularisation improves upon the original method.
-  They show empirically that their proposed methods performs better than most methods in particular MetaOPTNet
-  (I have not gone through the proofs of their theoretical claims and will base my opinion on this aspect from other more confident reviewers in this area)


Reason to reject/neutral the paper and areas that need clarification:

- Given that you have dropped the hyperparameter \lambda in your objective expression (You say you are learning the kernel so this is not necessary anymore), it seems to me that initialization of your networks seems crucial and could potentially make optimization very hard. Can you comment on this?

- The top of page 5 is very confusing and not very clear. I was wondering if you could give a more intuitive explanation of what is happening and why for example the implicit function theorem has to be used etc. I understand that this is to reduce complexity but a clearer explanation of this would be helpful.

- What is K in your paper? Is this the kernel that is learning through the feature maps? You define a K on the Z's. Have you tried to use a gaussian kernel on top of the deep kernel features instead of the just a linear kernel? You use the word RKHS and seems quite misleading (As they are usually associated with Infinite dim feature) if in reality all you are doing is deep kernel learning. This is not a problem perse just wanted this clarified as K is never properly defined (line 138-139)

-  Your Fig 1/2 is very misleading as you only show MAML papers such as [1] or [2] should be added. Else i do not see the point of this figure. Also the width is crucial in MAML and hasnt been properly studied

- Table 1/2/3 is missing [1, 2] which has very well documented code and beats MetaOptNet. Hence this comparison should not be omitted.

- Table 3: 5-shot, your method is not statistically significantly better than Proto-net and hence bolding your number is misleading



[1] : https://arxiv.org/pdf/1912.02738.pdf CODE: https://github.com/jinxu06/metafun-tensorflow
[2]: https://arxiv.org/pdf/2003.11539.pdf CODE: http://github.com/WangYueFt/rfs/

**Time Spent Reviewing:**

3

---

> ### Author Response · Authors · 2021-08-10
> **Reply to Reviewer S5bU (2/2)**
>
> **Q9.** "Table 3: 5-shot, your method is not statistically significantly better than Proto-net and hence bolding your number is misleading"
>
> - As suggested, we show that the improvement is statistically significant by increasing the number of testing tasks from 600 to 10,000. As shown in the table below, the improvement of MetaProx over ProtoNet is statistically significant (according to the pairwise t-test, with a p-value of 0.041). We will clarify it in the final version.
>
> ```
> Table: Accuracies (with 95% confidence interval) of 5-way few-shot
> classification on mini-ImageNet.
> --------------------------------------------------------------------------------
> method                5-shot (600 testing tasks)   5-shot (10000 testing tasks)
> --------------------------------------------------------------------------------
> ProtoNet                   68.2 ± 0.7                   68.23 ± 0.26
> MetaProx (proposed)        68.8 ± 0.8                   68.84 ± 0.29
> --------------------------------------------------------------------------------
> ```
>
> **Q10.** "The write up of the paper could be improved in particular the dual problem setup as well as their proposed computational tricks as mentioned above. Also in terms of the experiments i believe that methods mentioned above should be included for completeness or at least justified why they weren't added. I understand that the proposed method improves upon metaoptnet but that doesn't mean that it is necessarily the best method and hence should be compared to others as well."
>
> - Please see our replies above, which will be incorporated into the final version.

---

> > ### Comment · Reviewer_S5bU · 2021-08-23
> > **Thanks for the clarifications**
> >
> > I appreciate the authors extensive additional experiments and as well as additional clarifications on my confusions. All my main concerns have been lifted.
> > The addition of these experiments really strengthen the paper and hence i will increase my score to 6.

---

> ### Author Response · Authors · 2021-08-10
> **Reply to Reviewer S5bU (1/2)**
>
> Thanks for your review and suggestions. For the questions:
>
> **Q1.** "Given that you have dropped the hyperparameter $\lambda$ in your objective expression (You say you are learning the kernel so this is not necessary anymore), it seems to me that initialization of your networks seems crucial and could potentially make optimization very hard. Can you comment on this?"
>
> - Essentially, $\lambda$ is absorbed into the kernel. Empirically, network initialization is not critical. We simply use the default "_He Initialization_" in PyTorch.
>
> **Q2.** "The top of page 5 is very confusing and not very clear. I was wondering if you could give a more intuitive explanation of what is happening and why for example the implicit function theorem has to be used etc. I understand that this is to reduce complexity but a clearer explanation of this would be helpful."
>
> - This part shows how to obtain the two components in $\nabla_{(\theta, \phi)} \ell(\hat{y}, y)$ of line 154. Specifically, the first component $\nabla_{(\theta, \phi)} p$ can be directly computed. For $\nabla_{p} \alpha_{\tau}$, its computation is more complicated when the loss used is not square loss. Note that $p$ is input to the dual problem, and $\alpha_\tau$ is its solution. So, $\alpha_\tau$ depends implicitly on $p$, and we need to use the implicit function theorem to obtain $\nabla_{p} \alpha_{\tau}$. We will make this clearer in the final version.
>
> **Q3.** "What is K in your paper? Is this the kernel that is learning through the feature maps? You define a K on the Z's."
>
> - From line 138, $\mathcal{K}$ is the kernel of the RKHS $\mathcal{H}$. The model $f$ to be learned is in $\mathcal{H}$ (line 139), and from equation (4) just below line 139: $f$ is a function of $\mathbf{z}$, and so $\mathcal{K}$ is defined on $\mathbf{z}$'s (as can also be seen from line 144: $\mathcal{K}_{\mathbf{z}\_{i}} = \mathcal{K}(\mathbf{z}\_{i},\cdot) \in \mathcal{H}$).
>
> **Q4.** "Have you tried to use a gaussian kernel on top of the deep kernel features instead of the just a linear kernel? "
>
> - Previously, we have tried the Gaussian kernel $\mathcal{K}(\mathbf{x}_1, \mathbf{x}_2) = \exp \left(-\frac{\\|\mathbf{x}_1 - \mathbf{x}_2\\|^2}{2\sigma^2}\right)$ as suggested, and the results are not good. The table below shows its performance on the \textit{Sine} data in section 4.1, with $\sigma$ in the Gaussian kernel varying from $\\{0.01, 0.05, 0.1, 0.5, 1.0, 5.0\\}$. As can be seen, it is much worse than the linear kernel.
>
> ```
> Table: Average MSE (with 95% confidence intervals) of few-shot
> regression on the Sine (noise-free).
> -----------------------------------------------------------
>  kernel        2-shot             5-shot
> -----------------------------------------------------------
> RBF(0.01)    2.92 ± 0.19        2.78 ± 0.18
> RBF(0.05)    2.72 ± 0.18        2.36 ± 0.17
> RBF(0.1)     2.50 ± 0.17        2.25 ± 0.14
> RBF(0.5)     2.38 ± 0.16        1.71 ± 0.13
> RBF(1.0)     2.36 ± 0.16        1.68 ± 0.12
> RBF(5.0)     2.38 ± 0.15        1.72 ± 0.13
> linear       0.11 ± 0.01        0.01 ± 0.00
> -----------------------------------------------------------
> ```
>
> **Q5.** "You use the word RKHS and seems quite misleading (As they are usually associated with Infinite dim feature) if in reality all you are doing is deep kernel learning."
>
> - Indeed, RKHS can be finite-dimensional (e.g., RKHS of a polynomial kernel). For more details, please see, e.g., _A Primer on Reproducing Kernel Hilbert Spaces_, J.H. Manton and P.O. Amblard, 2015.
> - Note that indeed we are not performing deep "kernel learning". Specifically, the kernel $\mathcal{K}$ is given and fixed. The proposed MetaProx learns $f_\theta$, which is a function in the induced RKHS.
>
> **Q6.** "Your Fig 1/2 is very misleading as you only show MAML papers such as [1] or [2] should be added. Else i do not see the point of this figure."
>
> - Figure 1 aims to demonstrate that the learned initializations from MAML and MetaProx's $f_\theta$ are both close to the expected output (line 280), and thus can extract good meta-knowledge from the meta-training set.
> - To avoid clutterness, we only showed the classic algorithm MAML with MetaProx in Figures 1/2 of the submission. As suggested, we now add the curves of [1] and [2] in the figures ([https://www.dropbox.com/s/avc9a3vk81u85o0/figures_for_sine.pdf?dl=0](https://www.dropbox.com/s/avc9a3vk81u85o0/figures_for_sine.pdf?dl=0)) (the algorithm in [2] is originally designed for classification. To adapt it for regression, we pretrain the feature backbone on the meta-training set and learn an initialization for the output layer; at meta-testing, the output layer is finetuned on the testing task while keeping the backbone unchanged.)
> - Figure 1 (in the link above) shows that the initializations learned by MAML, MetaProx, and [2] are close to the expected output, while that from [1] is different.
> - Figures 2/3 (in the link above) show the regression results on two of the testing tasks ($\tau_1$ and $\tau_2$). As can be seen, MetaProx fits the target better. For task $\tau_1$ (Figures 2b/2c and Figures 3b/3c), the method in [2] fails to fit the target curves by fine-tuning the output layer. For task $\tau_2$ (Figures 2d/2e and Figures 3d/3e), all methods can fit the target curves but MetaProx is better (please refer to the tables in our reply to Q8 for the detailed numerical comparison on the meta-testing set).
>
> **Q7.** "Also the width is crucial in MAML and hasnt been properly studied":
>
> - As suggested, we run the proposed method use the setting in [1], and compare with the MAML results (reported in Table 1 of [1]) that use a larger (denoted LargeMAML) and wider network (denoted VeryWideMAML). As can be seen from the following Table, MetaProx has the best performance.
>
> ```
> Table: Average MSE (with 95% confidence intervals) of
> few-shot regression on the Sine using the settings in [1].
> ---------------------------------------------------------
>     method               5-shot            10-shot
> ---------------------------------------------------------
>  OriginalMAML         0.390 ± 0.156      0.114 ± 0.010
>    LargeMAML          0.208 ± 0.009      0.061 ± 0.004
>  VeryWideMAML         0.205 ± 0.013      0.059 ± 0.010
>     MetaFun[1]        0.040 ± 0.008      0.017 ± 0.005
> MetaProx (proposed)   0.010 ± 0.001      0.002 ± 0.001
> ---------------------------------------------------------
> ```
>
> **Q8.** "Table 1/2/3 is missing [1, 2] which has very well documented code and beats MetaOptNet. Hence this comparison should not be omitted."
>
> - As suggested, we add [1,2] as baselines for comparison.
> - The tables below show results for the regression experiments in section 4.1. As can be seen, MetaProx consistently outperforms MetaFun[1] and [2].
>
> ```
> Table 1: Average MSE (with 95% confidence intervals)
> of few-shot regression on QMUL (10-shot).
> --------------------------------------------------------------
>     method                in-range           out-of-range
> --------------------------------------------------------------
>     MetaFun[1]            0.022 ± 0.008      0.040 ± 0.009
>       [2]                 0.250 ± 0.015      0.312 ± 0.053
>   MetaProx (proposed)     0.012 ± 0.003      0.020 ± 0.005
> --------------------------------------------------------------
> ```
> ```
> Table 2: Average MSE (with 95% confidence intervals)
> of few-shot regression on the Sine and Sale datasets.
> ------------------------------------------------------------------------------
>                        sine (2-shot)    |    sine (5-shot)   |    Sale
>      method       ------------------------------------------------------------
>                      noise-free  noisy  |  noise-free  noisy | 1-shot  5-shot
> ------------------------------------------------------------------------------
>     MetaFun[1]         0.22      1.01   |     0.04     0.33  |   0.080  0.063
>         [2]            1.62      2.13   |     1.04     1.55  |   0.071  0.064
>   MetaProx (proposed)  0.11      0.43   |     0.01     0.13  |   0.061  0.060
> ------------------------------------------------------------------------------
> ```
>
> - For the few-shot classification experiment in section 4.2, [1,2] uses a different setting from ours. Specifically, [1] trains the meta-learner by combining the meta-training set and meta-validation set, and additional strategies (pretraining, augmentation, feature normalization) are used in [2]. We now add the following experiment with their settings and additional strategies. All methods use the ResNet-12 as the backbone. The table below shows the results. As can be seen, MetaProx again outperforms [1] and [2].
>
> ```
> Table 3: Accuracies (with 95% confidence interval) of 5-way few-shot
> classification on mini-ImageNet.
> -------------------------------------------------------------
>    method                     1-shot               5-shot
> -------------------------------------------------------------
> (use setting in [1])
>  MetaFun-DFP[1]             64.13 ± 0.13       80.82 ± 0.17
>  MetaFun-KFP[1]             63.39 ± 0.15       80.81 ± 0.10
> MetaProx (proposed)         64.96 ± 0.18       81.33 ± 0.13
> -------------------------------------------------------------
> (use setting in [2])
>  [2] (simple)               62.02 ± 0.63       79.64 ± 0.44
>  [2] (distill)              64.82 ± 0.60       82.14 ± 0.43
> MetaProx (proposed)(simple) 66.05 ± 0.21       82.83 ± 0.20
> -------------------------------------------------------------
> * additional strategies in [2]:
>     * simple  = pretraining + augmentation + feature normalization
>     * distill = simple + knowledge distillation
> ```

---

### Official Review · Reviewer_pmQA · 2021-07-14

**Rating:** 6
**Confidence:** 4

**Summary:**

The authors proposed a kernelized proximal regularization method (MetaProx) for meta learning. MetaProx combines deep kernel and proximal meta regularization, allowing learnable regularization by reformulating the proximal problem in dual space. Thorough theoretical analysis are provided as well as empirical studies on regression and classification task.

**Limitations And Societal Impact:**

There is no negative societal impact of this work.

**Main Review:**

Overall the paper is well written and easy to follow. The proofs are clear. The derived propositions supported the motivation of introducing learnable regularization.
My major concerns about this paper are as follow:
1. The author mentioned that the proposed method can introduce nonlinearity to the model other than CommonMean method. By using a linear kernel as base kernel, the only difference between MetaProx and CommonMean is that MetaProx utilizes nonlinear neural networks as embedding network while CommonMean doesn’t. And that’s also the reason it outperforms CommonMean. It makes the work looks a bit incremental from this perspective. Though I think the author made a good point about why it outperforms MetaOpt-SVM work.

2. In few-shot classification task, the original MetaOpt-SVM work mentioned that their method works better in large backbone network setting (e.g. ResNet12). However, in smaller backbone network setting, they do not have much advantages and hard to outperform ProtoNet. It might be fairer and more convincing to compare under different backbone setting to show consistent results.

**Time Spent Reviewing:**

5

---

> ### Author Response · Authors · 2021-08-10
> **Reply to Reviewer pmQA**
>
> Thanks for your review and suggestions. For the concerns:
>
> 1. "The author mentioned that the proposed method can introduce nonlinearity to the model other than CommonMean method. By using a linear kernel as base kernel, the only difference between MetaProx and CommonMean is that MetaProx utilizes nonlinear neural networks as embedding network while CommonMean doesn’t. And that’s also the reason it outperforms CommonMean. It makes the work looks a bit incremental from this perspective. Though I think the author made a good point about why it outperforms MetaOpt-SVM work."
>
> - A key novelty is on moving from the primal space in CommonMean to the dual space, which allows a significant reduction in complexity (line 133), and also naturally allows the extension to nonlinear meta-regularization (line 134). This would have been impossible when, for example, the cosine kernel is used (whose RKHS is infinite-dimensional). However, with the proposed formulation, it leads to the technical issues of how to obtain the learnable function $f_\tau$ and how to perform the meta-update efficiently. We showed how to compute the meta-gradients with a complexity that is linear in the number of meta-parameters and independent of the dimensionality of the induced RKHS. This is more efficient than meta-learning algorithms such as iMAML and MAML. Experimental results on regression and classification also verify the performance advantage of the learned kernelized regularizer.
>
> 2. "In few-shot classification task, the original MetaOpt-SVM work mentioned that their method works better in large backbone network setting (e.g. ResNet12). However, in smaller backbone network setting, they do not have much advantages and hard to outperform ProtoNet. It might be fairer and more convincing to compare under different backbone setting to show consistent results."
>
> - As suggested, we use ResNet-12 as the backbone for the few-shot classification experiment in Section 4.2. In the following table, result of MetaOptNet-SVM(cos) is obtained by running the code of MetaOptNet using the cosine kernel; results of the other baselines are from the MetaOptNet paper. As expected, the ResNet-12 backbone leads to better results than the shallow Conv-4 backbone across all methods. The proposed MetaProx still outperforms the baselines, and the improvement is statistically significant using the pairwise t-test at 95% significance level.
>
> ```
> Table: Accuracies (with 95% confidence interval) of 5-way few-shot
> classification on mini-ImageNet using ResNet-12.
> -------------------------------------------------------------
>      method                  1-shot              5-shot
> -------------------------------------------------------------
>  MetaOptNet-SVM(lin)      62.64 ± 0.61       78.63 ± 0.46
>  MetaOptNet-SVM(cos)      62.75 ± 0.42       78.68 ± 0.24
>        ProtoNet           59.25 ± 0.64       75.60 ± 0.48
>  MetaProx (proposed)      63.82 ± 0.23       79.12 ± 0.18
> -------------------------------------------------------------
> ```

---

> > ### Comment · Reviewer_pmQA · 2021-08-24
> > **Thanks for the authors' response**
> >
> > Thanks for the authors' detailed explanation.
> > Regarding the response of the first argument, I understand the author moved the problem from primal space to dual space to reduce computation complexity caused by RKHS( usually of infinite dimension). However, I think this is commonly used trick in kernel methods and thus weaken the novelty of this point.
> > Regarding the response of the second argument, I thank the author for taking time to provide additional experiments results and thus released my concern.
> > Overall, I'd like to raise my score to 6.

---

> > > ### Author Response · Authors · 2021-08-25
> > > **Thank you for increasing the score!**
> > >
> > > Thank you for increasing the score. For the additional comment:
> > >
> > > 1. "However, I think this is commonly used trick in kernel methods and thus weaken the novelty of this point."
> > >
> > > - In traditional machine learning, with kernel-induced feature mapping, sometimes one can still perform optimization in the primal space. In meta-regularization, the goal is to learn the regularizer. Without dual formulation, by the implicit function theorem, the complexity of computing the meta-gradient w.r.t. $f_\theta$ is cubic in the dimensionality of the induced RKHS. The proposed algorithm reduces the complexity to linear time by solving the inner problem in the dual space.

---

### Official Review · Reviewer_2wb5 · 2021-07-18

**Rating:** 7
**Confidence:** 3

**Summary:**

This paper proposes a new meta-learning algorithm. Following the meta-regularization method, this paper proposes a new proximal regularizer instead of the traditional regularizer to reduce the computational complexity significantly. The algorithm is reasonable and interesting which also shows high performance in experiments.

**Limitations And Societal Impact:**

It will be better to clarify the key motivation and implications to apply the proposed regularizer.

**Main Review:**

 Originality:

In meta-learning community, how to extract meta-knowledge for fast adaptation to a future task is the key problem. Metric-based algorithms learn meta-knowledge using a shared feature extractor and gradient-based meta-algorithms learn meta-knowledge by learning a common initialization. The above two branches draw a lot of attention these years, while meta-regularization, learning a common mean, is natural and reasonable but less popular. This paper discovers the potential of the meta-regularization algorithms.

This paper proposes a new meta-regularizer which is interesting and novel.

Quality:
pos: The idea is interesting and novel. The theoretical analysis and the experiments are solid.

cons: The motivation and implications of the new proposed regularizer are not very clear. Why we need to propose a new regularizer instead of the commonly used mean square loss?  Apart from the computational cost, is there any key difference between the proposed one and the traditional one?

**Time Spent Reviewing:**

1 hour

---

> ### Author Response · Authors · 2021-08-10
> **Reply to Reviewer 2wb5**
>
> Thanks for your review and suggestions. For the questions:
>
> 1. "The motivation and implications of the new proposed regularizer are not very clear. Why we need to propose a new regularizer instead of the commonly used mean square loss?"
>
> - We will make the motivation and implications in Section 1 clearer in the final version. Essentially, the new regularizer is motivated by the meta-regularizer in CommonMean, which has been shown to be effective. However, CommonMean is based on linear regression, and relies on its closed-form solution and uses matrix inversion to compute the meta-gradient. These become infeasible for nonlinear models (e.g., neural networks) and nonlinear losses (e.g., cross-entropy). On the other hand, the regularizers in recent meta-learning algorithms such as MetaOptNet and R2D2 are not learnable. These limitations motivate us to propose the kernelized proximal regularizer. Experimental results demonstrate its effectiveness.
>
> 2. "Apart from the computational cost, is there any key difference between the proposed one and the traditional one?"
>
> - CommonMean cannot be directly extended to nonlinear models and/or nonlinear losses, as the model no longer has a closed-form solution and inverting a possibly huge parameter matrix is infeasible. The proposed formulation uses the kernel trick and obtains the model efficiently in the dual space.

---

> ### Comment · Reviewer_2wb5 · 2021-08-26
> **Post-rebuttal**
>
> I have read all reviewers' comments and the authors' rebuttal, I still vote for acceptance and keep my score unchanged.

---

### Decision · Program_Chairs · 2021-09-27

**Decision:**

Accept (Poster)

**Comment:**

The paper initially received mixed ratings. Most concerns from the reviewers have been well addressed by the rebuttal and after the discussion period, all reviewers are now on the accept side. In particular, the reviewers have been convinced by the experimental results provided in the rebuttal. The area chair agrees with their assessment and follows their recommendation.

The final version of the paper should include these results and take into account the discussions that took place during the rebuttal period.